# Giant Viruses—Big Surprises

**DOI:** 10.3390/v11050404

**Published:** 2019-04-30

**Authors:** Nadav Brandes, Michal Linial

**Affiliations:** 1The Rachel and Selim Benin School of Computer Science and Engineering, The Hebrew University of Jerusalem, Jerusalem 91904, Israel; nadav.brandes@mail.huji.ac.il; 2Department of Biological Chemistry, The Alexander Silberman Institute of Life Sciences, The Hebrew University of Jerusalem, Jerusalem 91904, Israel

**Keywords:** Amebae viruses, viral evolution, protein domains, mimivirus, dsdna viruses, translation machinery, pandoravirus, NCLDV

## Abstract

Viruses are the most prevalent infectious agents, populating almost every ecosystem on earth. Most viruses carry only a handful of genes supporting their replication and the production of capsids. It came as a great surprise in 2003 when the first giant virus was discovered and found to have a >1 Mbp genome encoding almost a thousand proteins. Following this first discovery, dozens of giant virus strains across several viral families have been reported. Here, we provide an updated quantitative and qualitative view on giant viruses and elaborate on their shared and variable features. We review the complexity of giant viral proteomes, which include functions traditionally associated only with cellular organisms. These unprecedented functions include components of the translation machinery, DNA maintenance, and metabolic enzymes. We discuss the possible underlying evolutionary processes and mechanisms that might have shaped the diversity of giant viruses and their genomes, highlighting their remarkable capacity to hijack genes and genomic sequences from their hosts and environments. This leads us to examine prominent theories regarding the origin of giant viruses. Finally, we present the emerging ecological view of giant viruses, found across widespread habitats and ecological systems, with respect to the environment and human health.

## 1. Giant Viruses and the Viral World

Viruses are cell infecting agents present in almost every ecosystem. Questions the regarding viral origin and early evolution alongside all living organisms (bacteria, archaea and eukarya) are still wide open, and relevant theories remain speculative [1,2,3,4,5]. As viruses are exceptionally diverse and undergo rapid changes, it is impossible to construct an ancestral lineage tree for the viral world [6,7,8,9,10]. Instead, virus families are categorized according to the nature of their genetic material, mode of replication, pathogenicity, and structural properties [11].

At present, the viral world is represented by over 8,000 reference genomes [12]. The International Committee on Taxonomy of Viruses (ICTV) provides a universal virus taxonomical classification proposal that covers ~150 families and ~850 genera, with many viruses yet unclassified [13]. This collection provides a comprehensive, compact set of virus representatives.

Inspection of viral genomes reveals that most known viruses have genomes encoding only a few proteins. Actually, 69% of all known viruses have less than 10 proteins encoded in their genomes (Figure 1). It is a common assumption that viruses demonstrate near-optimal genome packing and information compression, presumably in order to maximize their replication rate, number of progenies, and other parameters that increase infectivity [14,15]. However, a debate is still ongoing over the generality of these phenomena [16], and there is a non-negligible percentage of larger viruses (Figure 1). On the far end of the distribution, there are viruses with hundreds of genes, most of them are considered giant viruses. While only 0.3% of the known viral proteomes contain 500 or more proteins, they encode as much as 7.5% of the total number of viral proteins (Figure 1B).

## 2. The Discovery of Giant Viruses

The first giant virus, *Acanthamoeba polyphaga mimivirus* (APMV), was discovered in 2003 [17]. Its size was unprecedented, being on the scale of small bacteria or archaea cells [18]. Unlike any previously identified virus, APMV could be seen with a light microscope [19,20]. Initially it was mistaken for a bacterium and recognized as a virus only ten years after its isolation [21]. Up to this day, most of its proteins remain uncharacterized [22,23]. Notably, even more than a decade after the discovery of APMV, the identification of giant viruses still sometimes involves confusion, as illustrated in the discovery of the *Pandoravirus inopinatum* [24], which was initially described as an endoparasitic organism, and *Pithovirus sibericum* [25], which was also misinterpreted as an archaeal endosymbiont (see discussion in References [21,26]).

In the following years after the initial discovery of APMV, many additional giant viral species have been identified and their genomes fully sequenced. Most giant viral genomes have been obtained from large-scale metagenomic sequencing projects covering aquatic ecosystems (e.g., oceans, pools, lakes and cooling wastewater units) [27,28]; others have been sequenced from samples extracted from underexplored geographical and ecological niches (e.g., the Amazon River, deep seas and forest soils) [29,30,31,32]. Despite the accumulation of many more giant virus representatives, the fraction of uncharacterized proteins in their proteomes remains exceptionally high [33]. Many of these uncharacterized proteins were also considered orphan genes (ORFans), i.e., no significant match to any other sequence was identified. For example, 93% of the *Pandoravirus salinus* proteins, the first representative of this family [34] were reported as ORFans.

At present, there are over a hundred giant virus isolates, which reveal fascinating and unexpected characteristics. These extreme instances on the viral landscape challenge the current theories on genome size and compactness in viruses, and provide a new perspective on the very concept of a virus and viral origin [4,20,28,35,36,37].

## 3. Definition of Giant Viruses

Attempts to distinguish giant viruses from other large viruses remain somewhat fuzzy [38,39]. Any definition for giant viruses would necessarily involve some arbitrary threshold, as virus size, whether physical, genomic or proteomic, is clearly a continuum (Figure 2). Giant viruses were initially defined by their physical size as allowing visibility by a light microscope [33]. In this report, we prefer a proteomic definition, even if somewhat arbitrary. We consider giant viruses as eukaryote-infecting viruses with at least 500 protein-coding genes (Figure 2). Of the 7,959 curated viral genomes (extracted from NCBI Taxonomy complete genomes), 24 represented genomes meet this threshold. Of these, we consider the 19 eukaryote-infecting viruses to be the giant virus representatives (Table 1), excluding the five bacteria-infecting viruses.

Recall that reported proteome sizes are primarily based on automatic bioinformatics tools, which may differ from the experimental expression measurements (e.g., mimivirus APMV [40]). Moreover, physical dimensions are not in perfect correlation with the number of proteins or genome size. For example, *Pithovirus sibericum*, which was recovered from a 30,000-year-old permafrost sample [25], is one of the largest viruses by its physical dimensions (1.5 µm in length and 0.5 µm in diameter). However, it is excluded from this report, as its genome encodes only 467 proteins.

## 4. Classification of Giant Viruses and the Question of Origin

All giant viruses belong to the superfamily of nucleocytoplasmic large DNA viruses (NCLDV), which was substantially expanded following the discoveries of giant viruses [41,42]. The NCLDV superfamily had traditionally been comprised of the following families: *Phycodnaviridae*, *Iridoviridae*, *Poxviridae*, *Asfarviridae* and *Ascoviridae* [43,44], for which a common ancestor had been proposed [45,46]. Following the inclusion of additional giant virus taxonomy groups (*Mimiviradae*, *Pandoravirus* and *Marseillevirus*) into the NCLDV superfamily, there remained only a handful of genes shared by the entire superfamily. Additional disparities in virion shapes and replication modes among NCLDV has led to the conclusion that the superfamily is not necessarily a taxonomic group, and that NCLDV families are more likely to have evolved separately [47,48,49].

Two models have been proposed for the evolution of giant viruses. According to the reductive model, an ancestral cellular genome became reduced in size, leading to the dependence of the resulting genome on host cells. The presence of genes carrying cellular functions in almost any giant virus (e.g., translation components) [50] is consistent with this model. An alternative and more accepted theory argues for an expansion model. According to this model, current giant viruses originated from smaller ancestral viruses carrying only a few dozens of genes, and through gene duplications and horizontal gene transfer (HGT), have rapidly expanded and diversified [48,51,52,53]. This model agrees with metagenomic studies and the wave of giant virus discoveries in recent years, suggesting massive gene exchange between giant viruses and a variety of organisms sharing the same ecosystems (e.g., Reference [32]).

Of special interest is the degree of similarity between giant viruses and their hosts. The amebae host in particular is often described as a melting pot for DNA exchange [54] that leads to chimeric genomes. The majority of genes in giant viruses and specifically *Mimiviridae,* have originated from the cells they parasitize (mostly amoeba and bacteria). Based on phylogenetic trees, it is likely that extensive HGT events have led to their chimeric genomes. It was also suggested that the spectrum of viral hosts may be larger than anticipated, including yet unknown species [55]. Therefore, comparative genomics of giant viruses infecting the same host is unlikely to unambiguously resolve questions of gene origin, namely, whether shared genes have originated from a common viral ancestor. Thus, the degree of similarity among giant viruses infecting different hosts is of special interest. For example, the phyletic relationship between *Mimiviridae* (which infect *Acanthamoeba*) and *Phycodnaviridae* (which infect algae) was investigated, and it was found that the algae-infecting *Chrysochromulina ericina* virus (CeV, Table 1) showed moderate resemblance to the amebae-infecting mimivirus [56]. As a result, suggestions were made to reclassify CeV as a new clade of *Mimiviridae,* rather than *Phycodnaviridae*. However, a later discovery of another algae-infecting *Phycodnaviridae* virus (*Heterosigma akashiwo* virus, HaV53) has provided a coherent phyletic relationship among *Phycodnaviridae*, thereby questioning this reclassification [52].

In summary, the taxonomy of giant viruses, like all viruses, is still unstable, and rapidly updated with new discoveries [31,57]. The origin and ancestry of giant viruses have remained controversial with questions of origin also unresolved [39]. Many newly discovered giant viruses are not compatible with the notion of a single common ancestor, as some giant viruses remain taxonomically isolated [4].

## 5. Common Features

Despite the ongoing debate on their origin, giant viruses still share some important features. All giant viruses belong to the dsDNA group, as do all NCLDV families. The total genome size of all the giant viruses listed in Table 1 is at least 288 Kbp (Figure 2). These giant viruses are classified into several families: *Mimiviridae*, *Pithoviridae*, *Pandoraviridae*, *Phycodnaviridae* and the *Mollivirus* genus [21,25].

All amoebae-infecting giant viruses rely on the non-specific phagocytosis by the amoebae host [55]. Interestingly, a necessary condition for phagocytosis is a minimal particle size (~0.6 μm [58]), as amebae (and related protozoa) naturally feed on bacteria. It is likely that this minimal size for inducing phagocytosis has become an evolutionary driving force for giant viruses. This fact, together with the largely uncharacterized genomic content of giant viruses, may suggest that much of the content in the genomes of giant viruses serves only for volume filling to increase their physical size.

Giant viruses share not only the cell entry process. When they exit the host cells during lysis, as many as 1000 virions are released from each lysed host via membrane fusion and active exocytosis [59], which are relatively rare exit mechanisms in viruses. 

Other than these genomic and cell-biology similarities, other features of giant viruses are mostly family-specific. For example, virion shapes and symmetries, nuclear involvement, duration of the infection cycle, and the stages of virion assembly—all substantially vary among giant viruses from different families [21,60,61].

## 6. Proteome Complexity and Functional Diversity

The majority of the giant virus proteomes remain with no known function (Figure 3). Actually, the fraction of uncharacterized proteins reaches 65–85% of all reported proteins in giant viral proteomes, many of them are ORFans. However, when proteomes of closely related species are considered, the fraction of ORFans obviously drops (by definition). For example, 93% of the proteins were reported as ORFans for the first representative of the *Pandoraviridae* family (*P. salinus*) [34]. But later, following the completion of five additional *Pandoravirus* proteomes (of the species *inopinatum*, *macleodensis*, *neocaledonia*, *dulcis*, and *quercus*), the number of *P. salinus* ORFans dropped to 29% (i.e., 71% of its genes now had a significant similarity to at least one other *Pandoravirus* protein sequence). Still, the vast majority of *Pandoravirus* proteins remain uncharacterized.

The most striking finding regarding the proteomes of giant viruses is the presence of protein functions that are among the hallmarks of cellular organisms, and are never detected in other viruses. To exemplify the complexity of proteome functions in giant viruses, we examined the proteome of the *Cafeteria roenbergensis* virus (CroV), which infects the marine plankton community in the Gulf of Mexico. *CroV* was sequenced in 2010 as the first algae-infecting virus in the *Mimiviridae* family. Unexpectedly, despite its affiliation with a recognized viral family, the majority of its proteins showed no significant similarity to any other known protein sequence. Of the remaining proteins that show significant basic local alignment search tool (BLAST) hits to other proteins from all domains of life, 45% are eukaryotic sequences, 22% are from bacteria, and the rest are mostly from other viruses, including other mimivirus strains. A similar partition of protein origin applies across other members of the *Mimiviridae* family [33].

The CroV proteome includes a rich set of genes involved in protein translation [62]. These genes include multiple translation factors, a dozen of ribosomal proteins, tRNA synthetases, and 22 sequences encoding five different tRNAs [62]. As a lack of translation potential is considered a hallmark of the virosphere, the presence of translation machinery components raised a debate on the very definition of viruses [63,64]. Similar findings were replicated in two other giant virus strains of the *Tupanvirus* genus in the same *Mimiviridae* family, which were recently isolated in Brazil [65]. The two strains have 20 open reading frames (ORFs) related to tRNA aminoacylation (aaRS), ~70 tRNA sequences decoding the majority of the codons, eight translation initiation factors, and elongation and release factors. The theory that translation optimization is an evolutionary driving force in viruses [66] may in part explain the curious presence of translation machinery in giant viruses. 

In addition to translation, numerous CroV proteins are associated with the transcription machinery. Specifically, the CroV proteome contains several subunits of the DNA-dependent RNA polymerase II, initiation, elongation, and termination factors, the mRNA capping enzyme, and a poly(A) polymerase. Presumably, the virus can activate its own transcription in the viral factory foci in the cytoplasm of its host cell [47].

Another unexpected function detected in CroV is DNA repair, specifically of UV radiation damage and base-excision repair. Other DNA-maintenance functions found in CroV include helicase and topoisomerases (type I and II), suggesting regulation of DNA replication, recombination and chromatin remodeling. 

Another rich set of functions related to protein maintenance include chaperons [67] and the ubiquitin-proteasome system [68]. Interestingly, some of these genes seem to be acquired from bacteria (e.g., a homolog of the Escherichia coli heat-shock chaperon). In addition, a rich collection of sugar-, lipid- and amino acid-related metabolic enzymes were also found [18,69], which occupy 13% of the CroV proteome (Figure 3).

It appears that the CroV proteome covers most functions traditionally attributed to cellular organisms, including: Protein translation, RNA maturation, DNA maintenance, proteostasis and metabolism. Although CroV exemplifies many widespread functions in giant viruses, each strain has its own unique functional composition. For example, the most abundant group of giant viruses in ocean metagenomes, the *Bodo saltans* virus (BsV), was recently identified and classified into the same microzooplankton-infecting *Mimiviridae* family [70]. Unlike the other family members, BsV does not have an elaborate translation apparatus or tRNA genes, but it carries proteins active in cell membrane trafficking and phagocytosis, yet more unprecedented functions discovered in viruses. 

## 7. Virophages and Defense Mechanism in Giant Viruses

Additional important players of genome dynamics in giant viruses are the virophages [72]. These are small double-stranded DNA (dsDNA) viruses that hitchhike the replication system of giant viruses following coinfection of the host, and are considered parasites of the coinfecting giant viruses [73]. Virophages (e.g., Sputnik 1–3, Zamilion) are associated with *Mimiviridae* representatives and their specific viral strain infectivity [21,74]. Additionally, short mobile genetic agents, called transpovirions (combining features of a transposon and a virion) [73], together with other mobile elements display complex ecological interactions with their hosts. Indeed, similar to eukaryotic transposons and endogenous viruses, sequences of a virophage (*Mavirus*) of *Cafeteria roenbergensis virus* (CroV), that were integrated to the genome of the protozoan host, serves an antiviral defense mechanism, which is activated by giant virus infection [75].

An even more unexpected finding is the discovery of a nucleic acid-based immunity in mimiviruses, resembling the adaptive (clustered regularly interspaced short palindromic repeats) CRISPR-Cas system in bacteria and archaea. Despite the differences to the canonical CRISPR-Cas system, an operon-like cluster of sequences derived from the *Zamilion* virophage was identified in mimivirus and experimentally validated to govern virophage coinfection. This cluster, coined MIMIVIRE, acts as a mimivirus virophage resistance element system with an exonuclease, helicase and RNase III identified in its vicinity [76]. The homology between the MIMIVIRE-associated exonuclease to the bacterial Cas-4 exonuclease was revealed by 3D protein structure analysis [77]. This CRISPR-Cas related function in mimiviruses are assumed to degrade foreign DNA, thereby constituting an antiviral innate immune system. It is likely that the CRISPR-Cas immune system in mimiviruses contributes to its sequence diversification as well by removing unessential host sequences. Alternative mechanisms that govern viral-host infection specificity and immunity may be discovered as our knowledge on virophages and other mobile elements is expending [72,78].

Altogether, a rich network of mobile genetic elements contributes to the host-virus coevolution and interviral gene transfer [78]. Virophages and other mobile elements could facilitate gene transfer, thereby having the potential to shape the genomes of giant viruses and impact their diversity [21,79,80].

## 8. The Emerging Ecological View

Viruses are the most abundant entities in nature. In marine and fresh water habitats, there are millions of viruses in each milliliter of water [81]. However, the collection of virus isolates is often sporadic, especially for those without clinical or agricultural relevance. The accelerated pace in the discovery of giant viruses reflects the increasing number of sequencing projects of exotic environments, including metagenomic projects [32,82].

Giant viruses have been isolated from numerous environmental niches and distant geographic locations, revealing their global distribution and diversity. Current evidence suggests that the representation of giant viruses is underexplored, especially in soil ecosystems [31] and unique ecological niches [83,84]. In fact, ~60% of the giant viral genomes were completed after 2013 (Table 1). Many more virus–host systems, most of them reported in the last five years, still await isolation, characterization and classification [84,85].

The hosts of contemporary isolates include mainly protozoa, specifically amoeba (Table 1). However, the prevalence of amoeba as hosts may in part be attributed to sampling bias, specifically to the widespread use of amoebal coculture methods for testing ecological environments [28,85]. 

Despite their prevalence, the impact of giant viruses on human health deserves further investigation [22]. Some initial reports show that APMV giant virus is able to replicate in human peripheral blood cells and to induce the interferon system [86]. Sequences of numerous giant viruses were identified as part of large-scale human gut microbiome sequencing projects [87], but their abundance, compositions and ecological roles are yet to be determined [88]. Reports are accumulating on the presence of giant viral sequences in human blood, as well as antibodies against giant viral proteins. Some reports associate mimirus and marseillevirus with a broad collection of human diseases (e.g., rheumatoid arthritis, adenitis, unexplained pneumonia, lymphoma), yet causal relationship is mostly missing [89]. The presence of giant viruses in almost any environment, including extreme niches and manmade sites (e.g., sewage and wastewater plants), suggests that the ecological role of these fascinating entities and their impact on human health are yet to be fully explored.

## Figures and Tables

**Figure 1 viruses-11-00404-f001:**
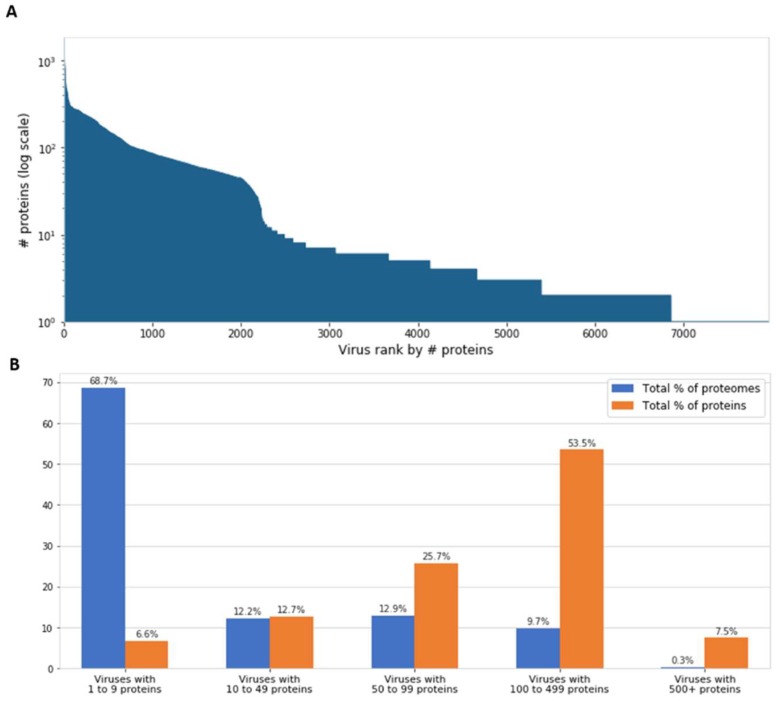
Number of proteins encoded by viruses. (**A**) The number of encoded proteins (*y*-axis) in all 7,959 viral representatives, ranked in descending order. (**B**) Partitioning of the 7959 viral proteomes by the number of encoded proteins. The 0.3% viral proteomes with the highest number of proteins (over 500) encode 7.5% of the total number of viral proteins.

**Figure 2 viruses-11-00404-f002:**
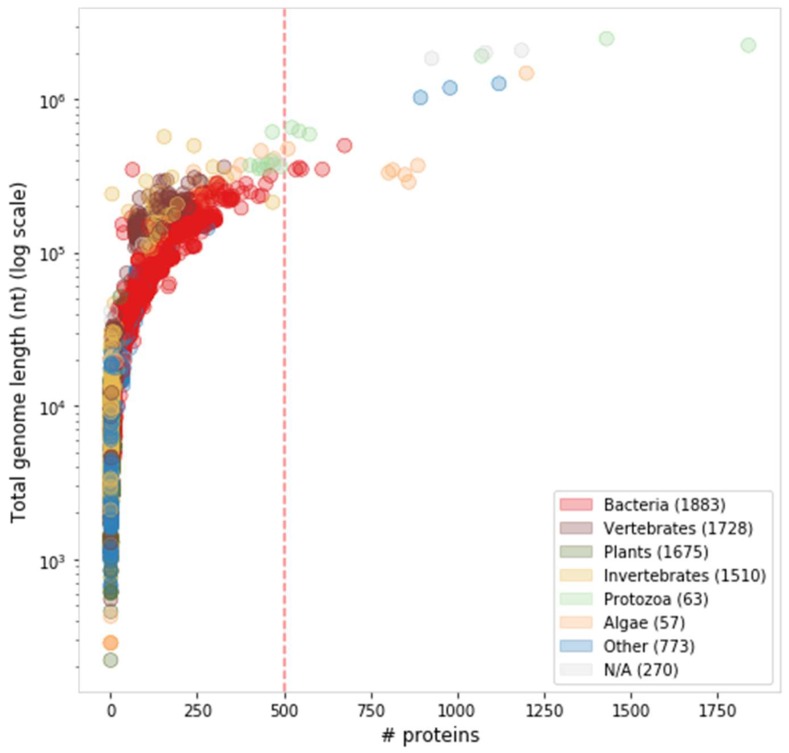
Distribution of viral proteome and genome sizes, colored by host taxonomy. There are 24 represented genomes that meet the threshold of ≥500 proteins (dashed red line), comprising five bacteria-infecting and 19 eukaryote-infecting viruses.

**Figure 3 viruses-11-00404-f003:**
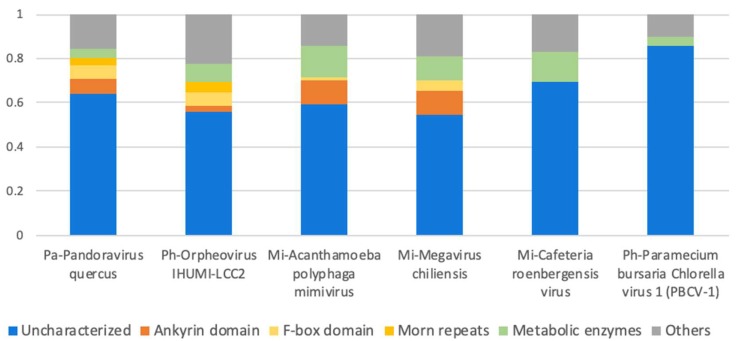
Protein function categories in six giant virus representatives from three families: *Mimiviridae* (Mi), *Pandoviridae* (Pa) and *Phycodnaviridae* (Ph). In all proteomes, the majority of proteins are uncharacterized. Short repeated domains such as ankyrin, F-box and MORM repeats are abundant in the proteomes of amebae-infecting giant viruses [71].

**Table 1 viruses-11-00404-t001:** Giant viruses.

Genome ^a^	Accession	Genome Length (kb)	# of Proteins	Host ^b^	Year ^c^
Mi-Acanthamoeba polyphaga mimivirus	NC_014649	1181.5	979	Pz, Ver	2010
Mi-Acanthamoeba polyphaga moumouvirus	NC_020104	1021.3	894	Pz, Ver	2013
Ph-Acanthocystis turfacea chlorella virus 1	NC_008724	288.0	860	Algae	2006
Mi-Cafeteria roenbergensis virus BV-PW1	NC_014637	617.5	544	Pz	2010
Pi-Cedratvirus A11	NC_032108	589.1	574	Pz	2016
Ph-Chrysochromulina ericina virus	NC_028094	473.6	512	Algae	2015
Mi-Megavirus chiliensis	NC_016072	1259.2	1120	Pz, Ver	2011
UC-Mollivirus sibericum	NC_027867	651.5	523	Pz	2015
Ph-Orpheovirus IHUMI-LCC2	NC_036594	1473.6	1199	Algae	2017
Pa-Pandoravirus dulcis	NC_021858	1908.5	1070	Pz	2013
Pa-Pandoravirus inopinatum	NC_026440	2243.1	1839	Pz	2015
Pa-Pandoravirus macleodensis	NC_037665	1838.3	926	Pz	2018
Pa-Pandoravirus neocaledonia	NC_037666	2003.2	1081	Pz	2018
Pa-Pandoravirus quercus	NC_037667	2077.3	1185	Pz	2018
Pa-Pandoravirus salinus	NC_022098	2473.9	1430	Pz	2013
Ph-Paramecium bursaria Chlorella virus 1	NC_000852	330.6	802	Algae	1995
Ph-Paramecium bursaria Chlorella virus AR158	NC_009899	344.7	814	Algae	2007
Ph-Paramecium bursaria Chlorella virus FR483	NC_008603	321.2	849	Algae	2006
Ph-Paramecium bursaria Chlorella virus NY2A	NC_009898	368.7	886	Algae	2007

^a^ Families: Mi, *Mimiviridae*; Ph, *Phycodnaviridae*; Pi, *Pithoviridae*; Pa, *Pandoraviridae*; UC, uncharacterized; ^b^ Pz, protozoa; Ver, vertebrates; ^c^ Year of genome submission to NCBI.

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
