# Peer review of "Giant Viruses—Big Surprises"

_viruses, 2019, doi:10.3390/v11050404_

Reviewer 1 Report

It is a review on giant viruses that is interesting, but poses some problems on data analysis. The debate on the source of genes of giant viruses has become a little too ideological, in particular it is difficult to think that genes of giant viruses originate from their host since about 80% of these genes are ORFans without an identified source, and it is difficult under these conditions to think that these genes have been acquired over time from their hosts.  This proposal is therefore the least plausible. Moreover, it has been demonstrated with the existence of mimivire that there is a CRISPR-like system in mimiviruses that makes it possible to get rid of alien sequences, and thus limits unnecessary sequence transfers into the genome of these giant viruses. Finally, genes that have some similarity to bacteria are in reality most of the time gene mosaics that include ORFans and genes from external sources, which is probably a widespread phenomenon in essential genes in giant viruses but also in cells. To conclude, the recent publication by Forterre’s team highlighted the existence of ribosomal proteins in a number of viruses and raises questions about the importance of translation in giant viruses, which is a subject that is still incomplete.

 Finally, the number of publications on the presence and isolation of viruses in pathological situations in humans is accumulating and cannot be just swept away because only one clinical microbiology team has so far focused on these giant viruses, which have attracted the attention of environmentalists. They have been evidenced in the blood, lymph nodes and respiratory system, with genome sequences proposed, and this deserves more than a line saying that this is questionable and to complete the bibliography.

Author Response

Referee 1:

It is a review on giant viruses that is interesting, but poses some problems on data analysis. The debate on the source of genes of giant viruses has become a little too ideological, in particular it is difficult to think that genes of giant viruses originate from their host since about 80% of these genes are ORFans without an identified source, and it is difficult under these conditions to think that these genes have been acquired over time from their hosts.  This proposal is therefore the least plausible.

 Reply: Thanks for the comments and suggestions.

Answering the question concerning the origin of genetic materials in giant viruses remains debatable and. We agree that a better definition is needed to clarify the statement “80% of the genes are ORFans”.  The referee correctly claims that with these values, it is unlikely that the host genome is the source.

We added a clarification in the revised manuscript for the definition of ORFans. We present ORFans in the case of Pandoravirus salinus (1,430 proteins), in view of other 5 additional Pandoravirus species (inopinatum, macleodensis, neocaledonia, dulcis, and quercus). Also, combined annotations on viral proteome resulting on somewhat reduced fraction of uncharacterized proteins. We mention that ORFans definition is fragile.

Moreover, it has been demonstrated with the existence of mimivire that there is a CRISPR-like system in mimiviruses that makes it possible to get rid of alien sequences, and thus limits unnecessary sequence transfers into the genome of these giant viruses. Finally, genes that have some similarity to bacteria are in reality most of the time gene mosaics that include ORFans and genes from external sources, which is probably a widespread phenomenon in essential genes in giant viruses but also in cells. To conclude, the recent publication by Forterre’s team highlighted the existence of ribosomal proteins in a number of viruses and raises questions about the importance of translation in giant viruses, which is a subject that is still incomplete.

 Reply: Thanks for the comments on the CRISPR like immunity in for giant viruses. Indeed, it is another ‘big surprise’ for giant viruses and a new way for cell ‘immunity’ The interesting observation regarding the ribosome relevant genes was mentioned quite extensively in our manuscript. We mention a review paper that thoroughly discuss this ‘translation relevance’/

We added the missing information and relevant references.

Finally, the number of publications on the presence and isolation of viruses in pathological situations in humans is accumulating and cannot be just swept away because only one clinical microbiology team has so far focused on these giant viruses, which have attracted the attention of environmentalists. They have been evidenced in the blood, lymph nodes and respiratory system, with genome sequences proposed, and this deserves more than a line saying that this is questionable and to complete the bibliography.

 Reply: We thank the referee for enlighten the relevant and importance of human health and giant viruses. We added an updated on human related observations as suggested with a few relevant references. Indeed we change the statement to mention that it is a topic that should be further developed.

Reviewer 2 Report

This manuscript is well written from the language point-of-view.

However, it dosnt describe anything new. All the facts are known and published. All the assumptions assumed in other papers.

In addition, there are inaccuracies along the paper.

I have written some of the points below:

Line 53-54 "The 0.3% viral proteomes with the highest number of proteins (over 500) encode 7.5% of the totalnumber of viral proteins."

This is a biased remark. The viruses are known for more than 120 years now while the giant viruses were discovered only 15 years ago. It is likely that many more giant viruses will be found and then the above remark will not hold.

Line 84-85 "we prefer a proteomic definition, even if somewhat arbitrary. We consider giant viruses as Eukaryote infecting viruses with at least 500 protein-coding genes (Figure 2)."

This statement is contradicted in the text itself:

In line 98-99 “5 bacteria-infecting and 19 eukaryote-infecting viruses.

Importantly, the natural hosts of the giant viruses are not known- Acanthamoeba polyphaga or others propagate many of the giant viruses but it is not certain at all that it is their host in nature.

Line 102-103 “All giant viruses belong to the superfamily of nucleocytoplasmic large DNA viruses (NCLDV),which was substantially expanded following the discoveries of giant viruses.

The wording nucleocytoplasmic large DNA viruses is not relevant as it includes the entire cell. The nucleus and the cytoplasmic part. So its everywhere.

Line 129 “coinfecting giant viruses [50]. These virophages play a fundamental role in accelerating the HGT process, including interviral gene.

The involvement of virophages in HGT was never shown. It was raised as an idea but no solid proof was ever give.

The reference number 52 (Boyer et al) is completely irrelevant in the context here.

Line 131-133 “ Additional agents that play a role in the rapid dynamics of giant viral genomes are a specific class of canonical transposable elements, which normally act in cellular organisms. They were recently shown to colonize Pandoravirus salinus [53].

Sun et al., 2015 that is cited is a bioinformatics paper and this whole theory is based on sequence alignment. The tole of the sequences that are found in Pandoravirus salinus was not shown in any experiment to be functional.

Line 134-136” It is estimated that the majority (50-85%) of genes in giant viruses have originated from the cells they parasitize [54]. Therefore, comparative genomics over giant viruses which infect the same host is unlikely to unambiguously resolve questions of gene origin, namely,

In reference 54 Moreira David., 2012 doesn’t mention any percentage of the genes in viruses that originate from eukaryotes. He says “as in the cases studied here for bacteria, the eukaryotic‐type genes present in viruses appear to be the result of horizontal transfers from their particular hosts (data not shown).”

And the number 50-85% is not there and anyway, none of this is even shown as data.

Line 147-150 “In summary, the taxonomy of giant viruses, as all viruses, is still very unstable, and rapidly updated with new discoveries [27]. The origin and ancestrally of giant viruses remained controversialwith questions of origin are also unresolved [34]. Many newly discovered giant viruses are not compatible with the notion of a single common ancestor, and some giant viruses remain completely undetermined [4].

This summary is known already from numerous papers.

Line 172 The whole paragraph) “Proteome complexity and functional diversity”

Nothing is new here -not a single idea/finding.

Author Response

This manuscript is well written from the language point-of-view. However, it dosnt describe anything new. All the facts are known and published. All the assumptions assumed in other papers. In addition, there are inaccuracies along the paper. I have written some of the points below:

Reply: We thank the referee for his comments and the effort to correct and refine our statement throughout.

Line 53-54 "The 0.3% viral proteomes with the highest number of proteins (over 500) encode 7.5% of the totalnumber of viral proteins."

This is a biased remark. The viruses are known for more than 120 years now while the giant viruses were discovered only 15 years ago. It is likely that many more giant viruses will be found and then the above remark will not hold

Reply: No question that we are only at the beginning of a wave of new giant discovery. The role of this assessment is mainly to discuss the current state of knowledge. Specifically, the fact that there is a huge gap between the apparently small number of such viruses and the large fraction they occupy.

Line 84-85 "we prefer a proteomic definition, even if somewhat arbitrary. We consider giant viruses as Eukaryote infecting viruses with at least 500 protein-coding genes (Figure 2)."

This statement is contradicted in the text itself:  In line 98-99 “5 bacteria-infecting and 19 eukaryote-infecting viruses. Importantly, the natural hosts of the giant viruses are not known- Acanthamoeba polyphaga or others propagate many of the giant viruses but it is not certain at all that it is their host in nature.

Reply: It is an arbitrary definition as stated. There is no contradiction in the reported numbers (24 total that is according to the cutoff of 500 proteins, only 19 that are eukaryote-infecting and are listed in our table). We refined the writing that indeed was confusion.

Line 102-103 “All giant viruses belong to the superfamily of nucleocytoplasmic large DNA viruses (NCLDV),which was substantially expanded following the discoveries of giant viruses.

 The wording nucleocytoplasmic large DNA viruses is not relevant as it includes the entire cell. The nucleus and the cytoplasmic part. So its everywhere.

Reply: Yes, we agree that this title is far from ideal. However, it is the common non-phyletic superfamily term that is used throughout.

Line 129 “coinfecting giant viruses [50]. These virophages play a fundamental role in accelerating the HGT process, including interviral gene. The involvement of virophages in HGT was never shown. It was raised as an idea but no solid proof was ever give.

The reference number 52 (Boyer et al) is completely irrelevant in the context here.

Reply: We removed the ref of Boyer at al. Providing supporting references to the potential role of virophages and other mobile genetic elements in altering the genome composition of these viruses. We revised the text to avoid misinterpretation.

Line 131-133 “ Additional agents that play a role in the rapid dynamics of giant viral genomes are a specific class of canonical transposable elements, which normally act in cellular organisms. They were recently shown to colonize Pandoravirus salinus [53].

 Sun et al., 2015 that is cited is a bioinformatics paper and this whole theory is based on sequence alignment. The tole of the sequences that are found in Pandoravirus salinus was not shown in any experiment to be functional.

Reply: We clarified the text to mentioned that it waits for functional testing. We remove the observation regarding the report on Pandoravirus salinus and the associated reference.

Line 134-136” It is estimated that the majority (50-85%) of genes in giant viruses have originated from the cells they parasitize [54]. Therefore, comparative genomics over giant viruses which infect the same host is unlikely to unambiguously resolve questions of gene origin, namely, In reference 54 Moreira David., 2012 doesn’t mention any percentage of the genes in viruses that originate from eukaryotes. He says “as in the cases studied here for bacteria, the eukaryotic‐type genes present in viruses appear to be the result of horizontal transfers from their particular hosts (data not shown).”

And the number 50-85% is not there and anyway, none of this is even shown as data.

Reply: We removed the numbers mentioned and we have refined the text to better represent the current state of the field. We also revised the supporting reference to these statements.

Line 147-150 “In summary, the taxonomy of giant viruses, as all viruses, is still very unstable, and rapidly updated with new discoveries [27]. The origin and ancestrally of giant viruses remained controversialwith questions of origin are also unresolved [34]. Many newly discovered giant viruses are not compatible with the notion of a single common ancestor, and some giant viruses remain completely undetermined [4].

This summary is known already from numerous papers.

 Line 172 The whole paragraph) “Proteome complexity and functional diversity”

Nothing is new here -not a single idea/finding.

Reply: This is the goal of the mini-review, accurate presentation of the current knowledge on giant viruses.

Reviewer 3 Report

This is a review on "giant viruses." In this manuscript, authors defined giant viruses as Eukaryote-infecting viruses with at least 500 protein-coding genes, and summarized their classification, common features, complexity and diversity, etc. This manuscript can give readers overviews of historical and recent topics of giant viruses, so it can be suitable for publication in Viruses as a review, after revisions as described below.

Major points:

(1) line 155: Authors described here about total genome size "at least 250 Kbp." However, in table 1, viruse which has a mimimum genome size is 288 kb. Why did authors write here "250"? What is the reason?

(2) line 156: Here authors did not describe about Phycodnaviridae, which is shown in Table 1. Instead of it, authors described about "Pithoviridae." Authors clearly defined giant viruses as Eukaryote-infecting viruses with at least 500 protein-conding genes in lines 84-85, so pithoviruses are excluded.  Nevertheless, why did authors included Pithoviridae, and excluded Phycodnaviridae here? Please describe more clearly and understandable for all readers. 

Minor points:

(1) There are several typo about "~viridae." Please proofread manuscript sufficiently. 

line 186: Mimiviradae must be corrected to "Mimiviridae."

line 192: Mimiviradae must be corrected to "Mimiviridae."

line 209: Mimiviradae must be corrected to "Mimiviridae."

               Pandoviradae must be corrected to "Pandoraviridae."

line 223: Mimiviradae must be corrected to "Mimiviridae."

(2) References 5 and 7 are identical.

Author Response

(1)   line 155: Authors described here about total genome size "at least 250 Kbp." However, in table 1, viruse which has a mimimum genome size is 288 kb. Why did authors write here "250"? What is the reason?

Reply: you are of course correct. We wanted to mention that the numbers are somewhat arbitrary (i.e 500 and not 499…).. To avoid confusion we corrected as suggested.

(2) line 156: Here authors did not describe about Phycodnaviridae, which is shown in Table 1. Instead of it, authors described about "Pithoviridae." Authors clearly defined giant viruses as Eukaryote-infecting viruses with at least 500 protein-conding genes in lines 84-85, so pithoviruses are excluded.  Nevertheless, why did authors included Pithoviridae, and excluded Phycodnaviridae here? Please describe more clearly and understandable for all readers. 

Reply: We mention in Table 1 one Pithoviridae representative (Cedratvirus) that meets the definition as stated. Other pithoviruses remained indeed outside of the list (as other Giants). By mistake that it was omitted in the text, we revised the text accordingly.

Minor points:

(1) There are several typo about "~viridae." Please proofread manuscript sufficiently. 

line 186: Mimiviradae must be corrected to "Mimiviridae."

line 192: Mimiviradae must be corrected to "Mimiviridae."

line 209: Mimiviradae must be corrected to "Mimiviridae."

               Pandoviradae must be corrected to "Pandoraviridae."

line 223: Mimiviradae must be corrected to "Mimiviridae."

(2)   References 5 and 7 are identical.

Reply: We thank the referee for careful reading. The typos were corrected throughout.

Round  2

Reviewer 1 Report

The current review is more balanced and I am in favor of this publication, however the authors have not integrated the notion of transpovirons, that of mimivires and the preventive role of mobile elements.

These three publications were however made in PNAS, Nature and Science. Indeed, transpovirons are the equivalent of transposons and are part of the mobile elements, mimivire is a system that is close to CRiSPR, the nuclease protein of the mimivire complex has been synthesized, crystallized and has an activity comparable to that of Cas-9. This explains why the probability that, like bacteria or archaea, unnecessary and imported host sequences are preserved is extremely unlikely, which is the very role discovered for CRiSPRs.

 Finally, CRISPR have shown an activity in the prevention of infection by virophages, highlighting the existence of a fight against mobile elements or alien nucleic acids, this must be clarified before this publication is accepted by me.

At least two recent reviews on giant viruses can be integrated into this reflection, one in Nature Microbiology Reviews and one in Frontiers, and this reviewer suggests that the authors read it to get a complete picture of exactly the same question they are asking themselves.

Author Response

Reply:

Thanks for the detailed clarifications.

The ‘story’ of mobile elements and the similarity to CRiSPR / CAS-9 is exciting. We added a short section (new section 5) to clarify this important addition as suggested. We also include the structural evidence of CAS nuclease and several overlooked references supporting the observations.

Due to the fact that Giant viruses are fairly ‘new’ to virology, several review articles summarizing ‘all that is known’ on Giant viruses were published. We added some of them to forward the reader to cover additional aspects of this evolving virology field.  

We hope that we still contribute to current knowledge. We mention that inconsistency in the definition may lead to confusion in the field, thus limiting our discussion to a proteomic-based threshold.

Actually, in this mini-review, we had not covered other aspects that were thoroughly discussed by other review papers (to avoid too much overlap) including the variations on the mode of replication, morphological aspects of the viruses, the geographical coverage and more. We added a set of recent review papers (~line 85) 

I thank the referee for his excellent comments and suggestions

Reviewer 2 Report

Unfortunately the paper "Giant Viruses-Big Surprises" still lacks any originality.

As mentioned in my previous review- the paper doesnt add any new ideas to the giant viruses young field, in my opinion.

Author Response

We appreciate your view, Still, based on a suggestion from the other referee, we develop recent discoveries of CRISPR-Cas immunity system and its relevance. We made an effort to shed light on critical aspects in this fast-evolving field and to put the field in the context of 'shared and unique features'. To avoid confusion we added several other review papers in the field to allow covering many of the topics that were not covered in this mini-review.